# Association between shell morphology of micro-land snails (genus *Plectostoma*) and their predator's predatory behaviour

Thor-Seng Liew and Menno Schilthuizen

Institute Biology Leiden, Leiden University, Leiden, The Netherlands
Naturalis Biodiversity Center, Leiden, The Netherlands
Institute for Tropical Biology and Conservation, Universiti Malaysia Sabah, Jalan UMS,
Kota Kinabalu, Sabah, Malaysia

## ABSTRACT

Predator–prey interactions are among the main ecological interactions that shape the diversity of biological form. In many cases, the evolution of the mollusc shell form is presumably driven by predation. However, the adaptive significance of several uncommon, yet striking, shell traits of land snails are still poorly known. These include the distorted coiled "tuba" and the protruded radial ribs that can be found in micro-landsnails of the genus *Plectostoma*. Here, we experimentally tested whether these shell traits may act as defensive adaptations against predators. We characterised and quantified the possible anti-predation behaviour and shell traits of *Plectostoma* snails both in terms of their properties and efficiencies in defending against the *Atopos* slug predatory strategies, namely, shell-apertural entry and shell-drilling. The results showed that *Atopos* slugs would first attack the snail by shell-apertural entry, and, should this fail, shift to the energetically more costly shell-drilling strategy. We found that the shell tuba of *Plectostoma* snails is an effective defensive trait against shell-apertural entry attack. None of the snail traits, such as resting behaviour, shell thickness, shell tuba shape, shell rib density and intensity can fully protect the snail from the slug's shell-drilling attack. However, these traits could increase the predation costs to the slug. Further analysis on the shell traits revealed that the lack of effectiveness in these anti-predation shell traits may be caused by a functional trade-off between shell traits under selection of two different predatory strategies.

## INTRODUCTION

Predator–prey interactions are among the key ecological interactions that shape the diversity of biological form (*Vermeij, 1987*). Predation may drive the evolution of prey morphology as prey forms that possess anti-predator characteristics increase survival and are selected under predation selection pressure. Among the studied prey traits, those of snail shells have been popular examples in demonstrating anti-predation adaptation (*Vermeij, 1993*). Among the reasons for this popularity are the fact that the shell is a conspicuous external structure, and the fact that its anti-predation properties may be

Corresponding author
Thor-Seng Liew,
thorsengliew@gmail.com

**Peer**J

observed directly as compared to other non-morphological anti-predation traits. Also, the interaction between predator and snail and the effectiveness of the anti-predation traits of the shell can be studied indirectly by examining traces and marks of both successful and unsuccessful predation on the shells (*Vermeij, 1982*; *Vermeij, 1993*). More importantly, the predator–prey interaction and evolution can be traced over time because shells with those predation marks are preserved in the fossil record (*Alexander & Dietl, 2003*; *Kelley & Hansen, 2003*).

The adaptive significance of shell anti-predation traits is better known for marine snails than for land snails (*Goodfriend, 1986*; *Vermeij, 1993*). This does not mean that land snails are less likely to be preyed upon in terrestrial ecosystems as compared to the marine ecosystems. In fact, the terrestrial ecosystem is a hostile environment to land snails, who face a taxonomically wide range of predators (*Barker, 2004* and reference therein). The fact that molluscs have diversified to become the second largest phylum on land after the arthropods (*Bieler, 1992*; *Brusca & Brusca, 2003*), suggests that land snails have evolved successful adaptations to deal with predation, and the evolution of shell morphology is likely to have played an important part.

The land snail shell is a single piece of coiled exoskeleton that consists of several layers of calcium carbonate. Its basic ontogeny follows a straightforward accretionary growth. Shell material is secreted by the mantle, which is located around the shell aperture, and is added to the existing aperture margin. Despite this general shell ontogeny that produces the basic coiled shell of all land snails, there is a great diversity of shell forms.

Many of the shell traits of land snails (e.g., whorl number and size, shell periphery form, umbilicus, shell coiling direction, aperture shape and size, and shell shape, thickness and size) are adaptive responses to abiotic ecological factors; by contrast, very few traits, viz. aperture shape and size, shell size, and shell wall thickness, are known to offer a selective advantage when faced with predation (*Goodfriend, 1986*). Since *Goodfriend*'s (*1986*) review, few additional studies have shown the adaptive significance of land snail shell traits under predation pressure, namely, aperture form (*Gittenberger, 1996*; *Quensen & Woodruff, 1997*; *Konumu & Chiba, 2007*; *Hoso & Hori, 2008*; *Hoso, 2012*; *Wada & Chiba, 2013*); shell form (*Quensen & Woodruff, 1997*; *Schilthuizen et al., 2006*; *Moreno-Rueda, 2009*; *Olson & Hearty, 2010*); shell ribs (*Quensen & Woodruff, 1997*); and shell coiling direction (*Hoso et al., 2010*).

Conspicuously lacking from this list are protruding radial ribs and distorted-coiling of the last whorl. These traits have been shown to have anti-predation function in marine snails (*Vermeij, 1993*; *Allmon, 2011*), but it remains unclear whether the same is true for land snails, where such traits are less common (*Vermeij & Covich, 1978*). Probably the only land snail taxon that possesses both of these traits is the genus *Plectostoma* (Fig. 1E). Some *Plectostoma* species have a regularly-coiled, dextral shell throughout their ontogeny, similar to most of the other gastropods. However, many *Plectostoma* species are unusual in having a shell that coils dextrally at the beginning of shell ontogeny (hereafter termed 'spire'), then changes direction at the transitional shell part (hereafter termed 'constriction'), and finally forms a last whorl that is detached from the spire and coils in an opposite

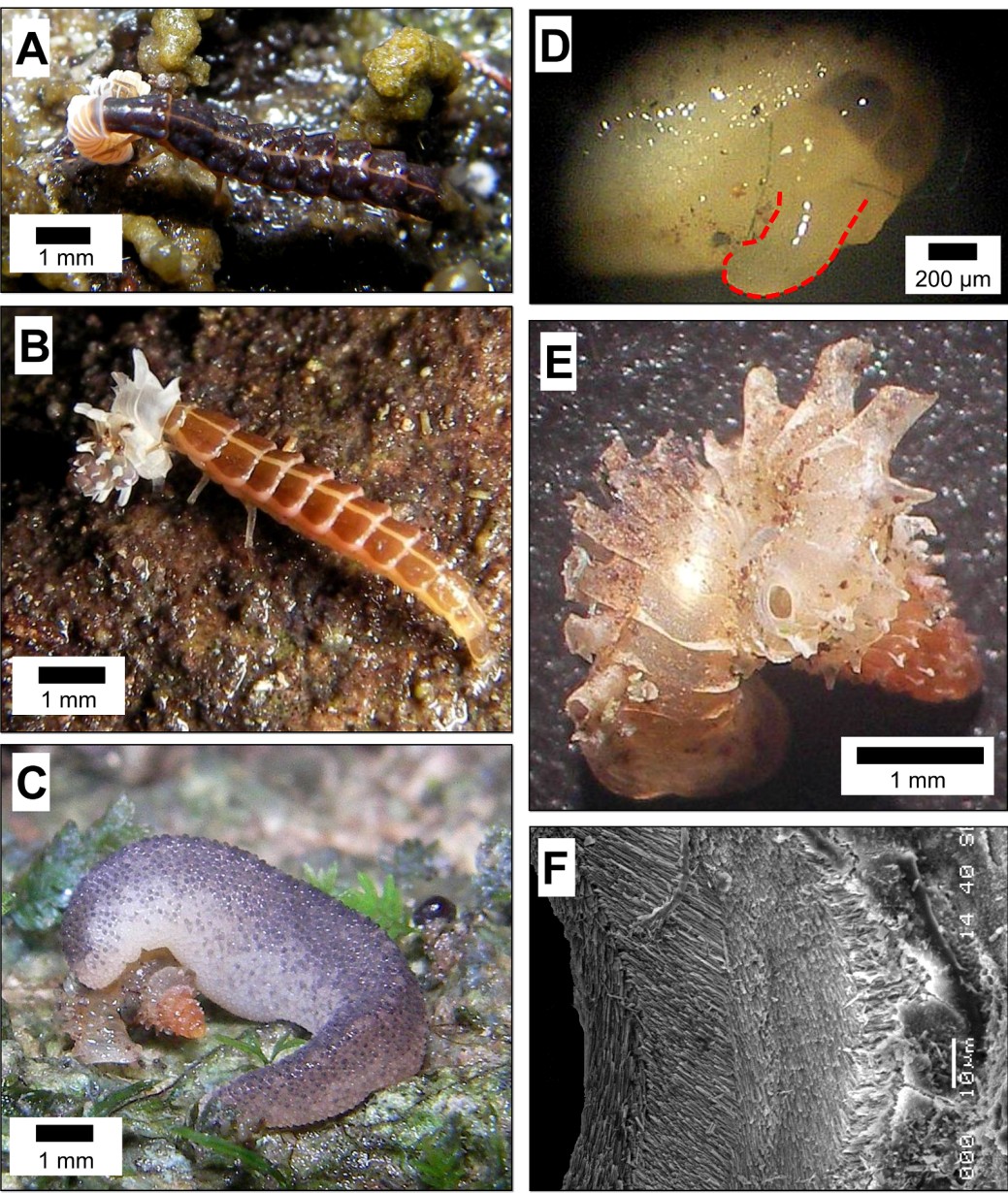

**Figure 1** **Predatory strategies that are used by *Atopos* slugs and Lampyridae beetle larvae to attack micro-land snails—*Plectostoma* species.** (A) *Pteroptyx* cf. *valida* (Olivier, 1883) larva, which was probably at its fifth instar, attacking *Plectostoma laidlawi* (Sykes, 1902) by shell-apertural entry. (B) *Pteroptyx tener* (Olivier, 1907) larva, which was probably at its fifth instar, attacking *Plectostoma fraternum* (Smith, 1905) by shell-apertural entry. (C) *Atopos* slug attacking *Plectostoma concinnum* (Fulton, 1901) by shell-drilling. (D) *Atopos* slug proboscis (marked with red outline) that was used for shell-drilling (the proboscis was not fully extended). (E) A drill hole on the shell of *Plectostoma concinnum* (Fulton, 1901) made by *Atopos*. (F) The appearance of the margin around the drill hole.

direction (hereafter termed 'tuba'; *van Benthem-Jutting, 1952*; *Vermeulen, 1994*; *Liew et al., 2014a*). Similar morphological transitions during shell ontogeny are known for other extant and fossil molluscs (e.g., *Okamoto, 1988*; *Clements et al., 2008*). In addition to this irregular coiling, there is great diversity in the shell radial ribs of *Plectostoma* in terms of density, shape, and intensity (i.e., amount of shell material in the ribs) (*van Benthem-Jutting, 1952*; *Vermeulen, 1994*). Clearly, *Plectostoma* is a good model taxon to improve our understanding of the ecological function of both of these unusual shell traits.

So far, the only known predator of *Plectostoma* snails is the slug *Atopos* (Rathousiidae) that uses a shell-apertural entry strategy to attack juvenile snails or uses a shell-drilling strategy to attack adult snails (*Schilthuizen et al., 2006*; *Schilthuizen & Liew, 2008*). In addition, we have also observed *Pteropyyx* beetle larvae (Lampyridae) attacking *Plectostoma* snails using a shell-apertural entry strategy (Figs. 1A and 1B; File S1, Page 1: Table S1). It has been suggested that predatory behaviour within a taxon would be quite conserved (*Barker, 2004* and reference therein).

Indeed, the predatory behaviour of these two predator taxa are generally concordant with that recorded from previous studies. Lampyridae beetle larvae use shell-apertural entry to attack and consume the snail (*Clench & Jacobson, 1968*; *Thornton, 1997*:65; *Archangelsky & Branham, 1998*; *Wang et al., 2007*; *Madruga Rios & Hernández Quinta, 2010*, for details see File S1, Page 3: Table S2). Rathousiidae slugs are known to have two strategies to attack and to consume the snail. Primarily, it uses shell-apertural entry (*Heude, 1882–1890*; *Kurozumi, 1985*; *Wu et al., 2006*; *Tan & Chan, 2009*) and secondarily, it uses shell-drilling when the opening of the prey is not available or accessible (*Kurozumi, 1985*; *Wu et al., 2006*; for details see File S1, Page 4: Table S3).

Although some of the *Plectostoma* shell traits have been shown to have some association with the shell drilling behaviour of the rathousiid slug *Atopos* (*Schilthuizen et al., 2006*), it is unclear how exactly the shell traits help *Plectostoma* defend against attacks from the *Atopos* slug and *Pteroptyx* larva. Direct observations and experiments on the interaction between the *Plectostoma* snails and their predators are prohibited by the predators' ecology. Both are nocturnal predators and they probably hide in the cracks of limestone rocks during the day. Hence, they appear to be very sensitive to light and manipulation.

Here, we attempt to reconstruct the predatory strategies of one of the predators, the *Atopos* slug, against the *Plectostoma* snail and try to empirically unravel any anti-predation function of the unusual *Plectostoma* shell traits through a series of experiments, and direct and indirect observations (hereafter known as "Tests"). We examined the effectiveness of several *Plectostoma* shell traits, namely, (1) ribs on shell surface ; (2) shell whorl thickness; (3) shell tuba; and (4) snail resting behaviour. These three shell traits and one behavioural trait were selected because these are known in other snail taxa for having antipredation properties against shell-apertural entry and shell-drilling behaviour by other predators (see overview in *Goodfriend, 1986*; *Vermeij, 1993*). We examined the effectiveness of the first three shell traits of *Plectostoma* against *Atopos* slug shell-drilling (Test 1); and the effectiveness of the last two traits of *Plectostoma* against *Atopos* slug shell-apertural entry (Test 2). Additionally, we investigate possible constraints in the development of
anti-predation shell traits. Finally, we discuss the results of this study in the context of predator–prey interactions and shell-trait evolution in general.

## MATERIALS AND METHODS

### Ethics statement

The permissions for the work in the study sites were given by the Wildlife Department of Sabah (JHL.600-6/1 JLD.6, JHL.6000.6/1/2 JLD.8) and the Economic Planning Unit, Malaysia (UPE: 40/200/19/2524).

### Predation tests

#### Test 1: Plectostoma snails' anti-predation traits against Atopos slug shell-drilling behaviour

Study on predatory drill holes on the shell provide information about the predator's drilling behaviour (*Kowalewski, Dulai & Fürsich, 1998*). *Atopos* and other Rathousiidae slugs drill a distinctive hole in the prey shells (*Schilthuizen et al., 2006*; Figs. 1E and 1F; File S1, Page 2; *Kurozumi, 1985*; *Wu et al., 2006*). Thus, the location and the size of the drill hole provide important information about the drilling behaviour of the slugs. In test 1, we tested the effectiveness of the tuba and shell ribs by examining whether *Atopos* drill holes on the tuba of the prey shell (Test 1a) and whether *Atopos* have a tendency for drilling holes between two ribs (Test 1a). In addition, we also examined the correlation between the rib density and other shell traits, such as rib intensity (i.e., amount of shell material in the ribs) (Test 1b), shell whorl thickness (Test 1c), and shell size (Test 1c) (Fig. 2).

#### Test 1 (a)—Association between slug shell-drilling, and adult snail shell tuba and rib density

Like in marine predator-snail interactions, where predators tend to drill a hole at less-ornamented positions of the prey shell (*Kelley & Hansen, 2003*) we may expect *Atopos* to drill its holes preferentially between shell ribs, rather than through them. Conversely, if snail shell ribs are adaptive traits in the context of the slug's shell-drilling behaviour, we would expect the snail shell to have evolved more densely-placed, thicker, and more protruded ribs to defend themselves against shell drilling predators.

To examine the association between shell rib density and drill hole position, we studied *Plectostoma* shell specimens from museum collections collected from two limestone outcrop, namely, Batu Kampung (5°32′11″N 118°12′47″E), and Batu Tomanggong Besar (5°32′3″N 118°23′1″E). These two limestone outcrops support dense *Plectostoma* populations, which show high variability in shell rib density. We selected museum specimens that belongs to two samples (i.e., populations) from Batu Kampung (*P. concinnum*, collection numbers BOR 1690, BOR 2196), and 9 samples (i.e., populations) from Batu Tomanggong Besar (collection numbers RMNH.MOL 330506; *P.* cf. *inornatum*: Samples T29, T33, T34, and T45; *P. fraternum*: Samples T7, T21, T22, and T42; and *P.* cf. *fraternum*: Sample T 44). All were collected between April 2002 and January 2004.

Each of the samples consisted of *Plectostoma* empty shells collected beneath the rock face where living *Plectostoma* individuals were also found. For each sample, shells with

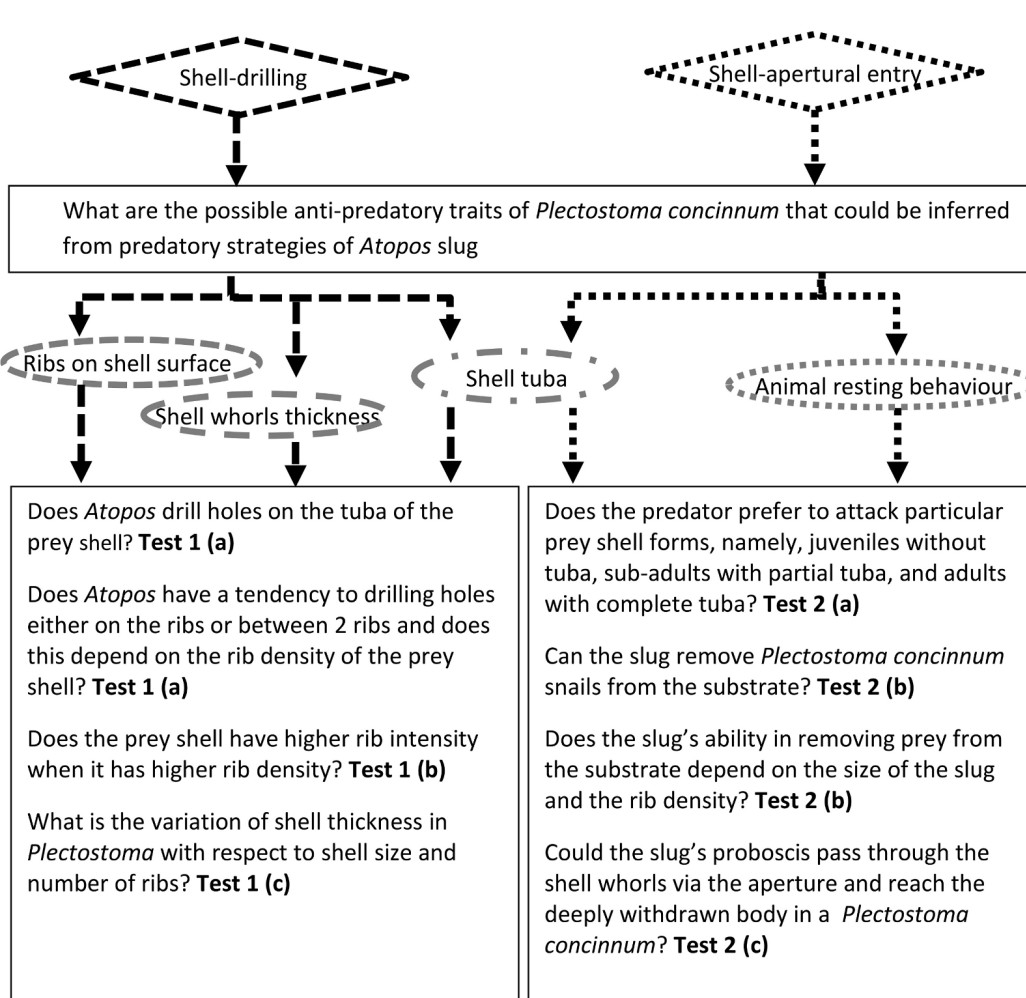

**Figure 2 Flowchart shows experimental design for 8 research questions of this study.** Bold text represents the respective tests for each research question; text bounded in each diamond shape represents the predatory behaviour of *Atopos*; text bounded in each oval shape represents the *Plectostoma* shell trait that was tested for their anti-predation property.

a characteristic *Atopos* drill hole were selected for analysis. We divided the shells into two groups based on the drill hole position: (1) hole directly through the shell wall and located between two ribs (hereafter BETWEEN RIBS), and (2) hole drilled through one or two ribs as well as the shell wall (hereafter ON RIBS). The two groups were used as the dependent variable, and were binary scored as (1) for BETWEEN RIBS and (0) for ON RIBS. In addition, we identified three predictor variables that may influence the slug drilling behaviour. First, the slug proboscis size, which was measured as the greatest diameter (mm) for circular and slightly oval drill holes (hereafter HOLE SIZE). Second, the rib density of the shell which was quantified as the total number of ribs on the shell (hereafter RIB DENSITY) because all shells have a similar number of whorls (mean: 5.15, SD: 0.35; File S2, Page 22: Table S2). Lastly, the random chance—the probability that a hole was made in between ribs, which is related to the HOLE SIZE and RIB DENSITY.

For example, by random chance, a slug with a narrow proboscis (i.e., low HOLE SIZE) has a greater probability to drill a hole in between the ribs on a shell that has fewer ribs (low RIBS DENSITY) because more rib spacings that are larger than the slug proboscis size are available. Thus, we counted total number of rib spacings larger than HOLE SIZE (hereafter CHANCES).

We used a logistic regression to model the likelihood that the slug drills a hole either BETWEEN RIBS or ON RIBS as a function of HOLE SIZE, RIB DENSITY, and CHANCES (i.e., Predicted logit of (BETWEEN RIBS) $= \beta_0 + \beta_1^*$ (HOLE SIZE) $+ \beta_2^*$ (RIB DENSITY) $+ \beta_3^*$ (CHANCES). Our objective was to investigate the amounts of variance attributable to each predictor variable. The analysis was done in R statistical package 2.15.1 (R Core Team, 2012) and the R scripts can be found in File S3.

### Test 1 (b)—Correlation between Plectostoma shell rib density and rib intensity

In addition to rib density, it is essential to quantify the amount of shell material that *Plectostoma* snails invest to grow thick and protruded ribs (hereafter we call this rib intensity). However, we cannot quantify this from the same shell remains that we had used in test 1(a) because the shell ribs of these specimens were heavily eroded. Thus, we analysed rib intensity from 14 preserved *Plectostoma* individuals that were collected alive from the same rock face at Batu Kampung and Tomanggong Besar, where the shell remains were collected (collection number RMNH 330508; T 21 ($n = 3$), T 22 ($n = 1$), T 42 ($n = 2$), T 7 ($n = 1$), T 44 ($n = 1$), BOR 2991 ($n = 3$), T 33 ($n = 3$)). These 14 shells have different rib densities (47–138 ribs per shell), which spans the broadest possible range of rib density, and have the most intact ribs on the shell.

We used X-ray microtomography (μCT) to estimate the amount of shell material that *Plectostoma* invests in rib growth (Fig. 3). First, we obtained a series of X-ray tomographies of each shell with a high-resolution SkyScan 1172 (Aartselaar, Belgium). The scan conditions were: 60 kV; pixels: 668 rows $\times$ 1000 columns; camera binning $4 \times 4$; image pixel size 7–9 μm; rotation step 0.5°; rotation 360° (Step 1 in Fig. 3).

Then, we reconstructed 2D grey scale images (i.e., cross-sections) from X-ray tomography series with NRecon 1.66 (©SkyScan). The settings were: beam-hardening correction 100% and ring artifacts reduction 20. Next, these 2D images were transformed to the final half-tone binary images for each shell in CTAnalyser 1.12 (©SkyScan). This was done by filtering out grayscale index <70. At this stage, each shell was represented by hundreds of 2D cross-section binary images (Step 2 in Fig. 3).

Each of these 2D images consisted of white and black pixels, where the white pixels represent the solid shell material (shell together with ribs) and the black pixels are background or lumen. When the series of cross-section images was analysed, the total voxels which represent the shell material volume could be determined. Hence, we analysed the volume of shell material from two datasets of each shell. The first was the original 2D cross-section binary images which represent the total volume of shell material contained in whorls and ribs (Step 3 in Fig. 3). The second was the volume of shell material contained in the shell whorls only, after removal of the shell ribs from each cross-section image.

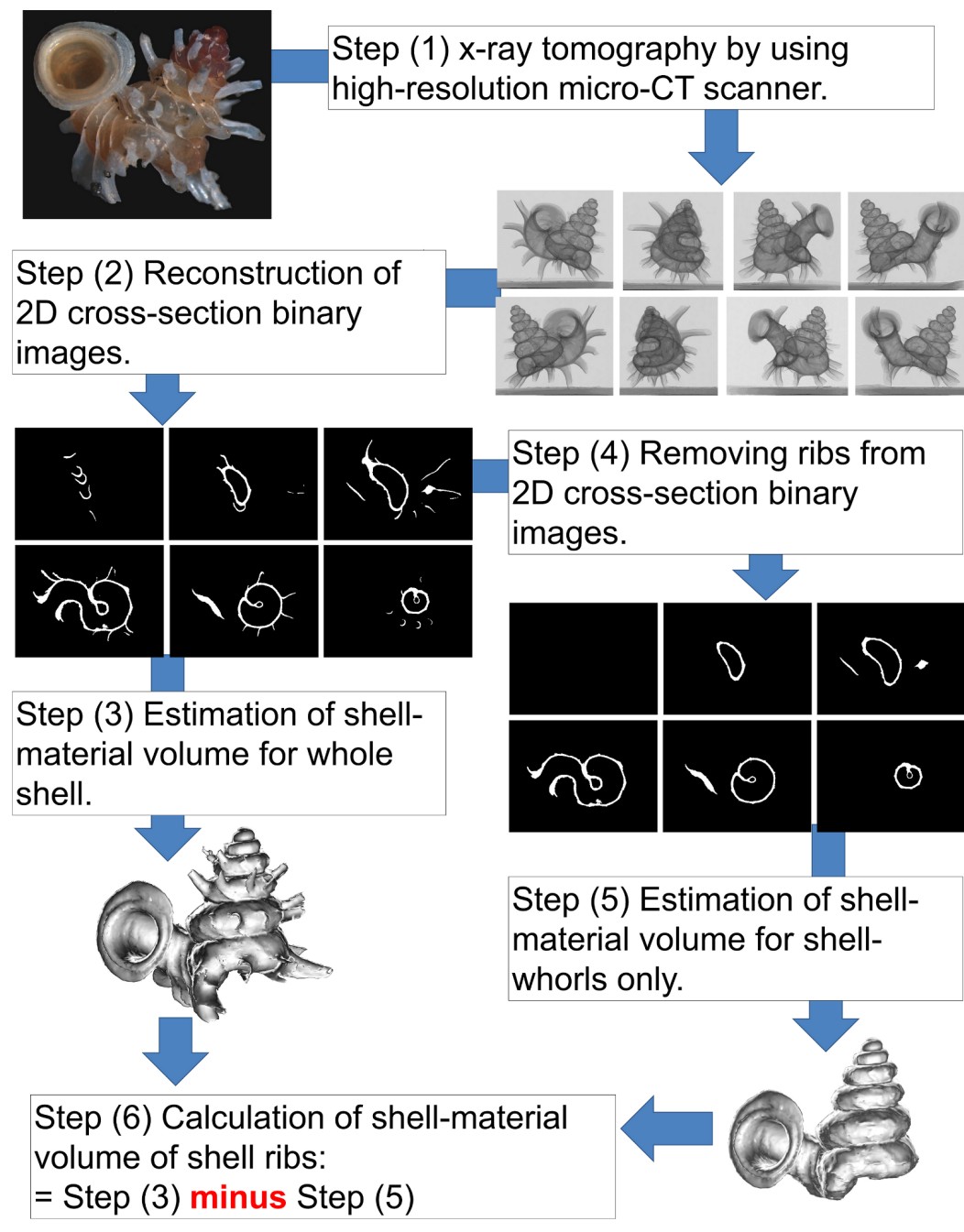

Figure 3 Procedures used to quantify the shell volume of material of the ribs and shell whorls (Test 1b).

The latter was done manually by changing white rib-pixels into black ones in Paint (©Microsoft Windows 7) (Step 4 in Fig. 3). After that, the volume of shell material was calculated for both datasets with Individual 3D object analysis, as implemented in CT Analyser 1.12 (©SkyScan) (Step 5 in Fig. 3). Finally, the rib intensity (i.e., amount of shell material in the ribs) was calculated by subtracting the volume after rib removal from the total volume with ribs included (Step 6 Fig. 3).

We wished to test if there is a significant correlation between rib intensity and number of ribs. However, as there is variability in the shell size for the shells that vary in rib density, we quantified a set of size variables of the shell (number of whorls, height, width, and volume of shell material of the shell whorls after rib removal) and then checked for confounding effects of shell size variables with the anti-predation shell traits. The results showed that only one of the shell size variables, i.e., the volume of shell material after rib removal, is significantly correlated with the anti-predation shell traits (File S2, Page 23: Table S3).

So, we also ran an additional partial correlation test between the same two variables (rib intensity vs. number of ribs) after controlling for total volume of shell material after rib removal, to account for confounding effects of the shell size difference. Pearson correlations were performed in the two tests as all variables were normally distributed (Shapiro–Wilk normality test, $p > 0.05$) with R statistical package 2.15.1 (*R Core Team, 2012*) and R scripts can be found in File S3.

### Test 1 (c)—Relationships between shell thickness, rib number, and shell size

We obtained 3D models (PLY format) of each of the 14 shells by using the original 2D cross-section binary images that were obtained from experiment 1(b). After that, we measured the shell thickness of the last spire whorl by making a cross-section of the digital 3D models with Blender 2.63 (Blender Foundation, www.blender.org). We obtained the shell thickness data from the digital 3D models instead of the actual specimens because it is difficult to make a clean cross-section on this tiny shell.

In order to assess if the prey invests more shell material in increasing the shell thickness when it invests less in the ribs, we tested the correlation between shell thickness and number of ribs. Similar to test 1(b), we also ran an additional partial correlation test between the same two variables after controlling for the volume of shell material after rib removal, to account for the variability in shell size differences. In addition, the relationships between shell thickness, rib number, and shell size were explored. Pearson correlations were performed in these tests as all variables were normally distributed (Shapiro–Wilk normality test, $p > 0.05$) in R statistical package 2.15.1 (*R Core Team, 2012*) and R scripts can be found in File S3.

### Test 2: Plectostoma snails' anti-predation traits against the apertural-entry behaviour of the Atopos slug

Unlike test 1, testing the associations between the prey shell traits and slug apertural-entry behaviours is more challenging because this type does not leave a distinctive trace on the prey shell after successful predation. One of the ways to assess the interaction between the prey shell and predator behaviour is with a manipulative experiment. However, this slug is very sensitive and hard to manipulate and thus sufficient replicates cannot be achieved. Hence, we used observations (Test 2a), indirect data (Test 2b), and a simulative model (Test 2c) to unravel the effectiveness of the shell traits against the predator shell-apertural entry behaviour (Fig. 2).

### Test 2 (a)—Observation of predator preferences for three different prey shell forms

So far, we have not observed drill holes in *Plectostoma* shells with no tuba (*Schilthuizen et al., 2006*; Liew T-S, unpublished data, 2013), and only once the slug was seen attacking a juvenile prey without a tuba by shell-apertural entry (*Schilthuizen & Liew, 2008*). However, we do not know if the slug has a preference for juvenile or adult prey. Thus, we conducted a small experiment to check prey age preference.

Two *Atopos* slugs, with body lengths of 7 and 15 mm, were collected from Site A (No. 7 & 8 in File S1, Page 1: Table S1). Each of the slugs was kept in a plastic box (12 cm × 8 cm × 7.5 cm), which contained a piece of limestone rock and its temperature and humidity were controlled. The boxes were kept under the table in a room with opened window to simulate the natural habitat for the slugs that are active nocturnally and rest in a shaded place during the daytime.

Live *P. concinnum* individuals were collected from Batu Kampung for this test. For each experiment, three individuals were placed on the rock in the plastic boxes. The three preys represented three different shell forms (i.e., growth stages): (1) shell with no tuba and peristome lip (juvenile, e.g., Fig. 4A: shells e–g), (2) shell with partial tuba but no peristome lip (sub-adult, e.g., Fig. 4A: shells h–j), and (3) shell with fully grown tuba and peristome lip (adult, e.g., Fig. 4A: shell l). During the experiment, the interactions between predator and prey were checked every 3 h to minimise the disturbance to the organisms. Each experiment ended after the slug was observed inactive (i.e., hiding under the rock) and at least one of the prey was consumed. After that, the three prey shells were removed for further analysis, and replaced with another three living snails to start a new experiment.

We ran nine such experiments, one with slug No.7 and eight with slug No. 8. After each experiment, each of the three shell forms was scored as having either survived or died (Specimens deposited in BOR 5657). Also, the shell of each dead prey was examined for possible traces left by slug predation. In addition, we also estimated the predator's attack and consuming time from the time intervals between the moments when all prey were last seen alive and the moment the experiment was ended. After each experiment, we checked if all three shell forms were equally likely to be killed by the predator.

### Test 2 (b)—Effectiveness of resting behaviour of Plectostoma snails against Atopos shell-apertural entry predatory behaviour

When a *Plectostoma* snail is resting or is disturbed, it withdraws its soft body into the shell and adheres its shell aperture to the substrate. Thus, when the snail is in this position, its aperture is not accessible to the slug, and for the slug to access the shell aperture, it would need to remove the shell from the substrate. In this test, the ability of the slug to manipulate the adherent prey shell was inferred by examining the drill hole location of the specimens used in Test 1(b). We predict that the sector of the shell facing the substrate is less susceptible to drilling by the slug if it is unable remove the adherent prey shell from the substrate.

For each of 133 shells, we recorded the location of the drill hole. We divided drill-hole locations of these shells into four categories, which represent different sectors, namely:

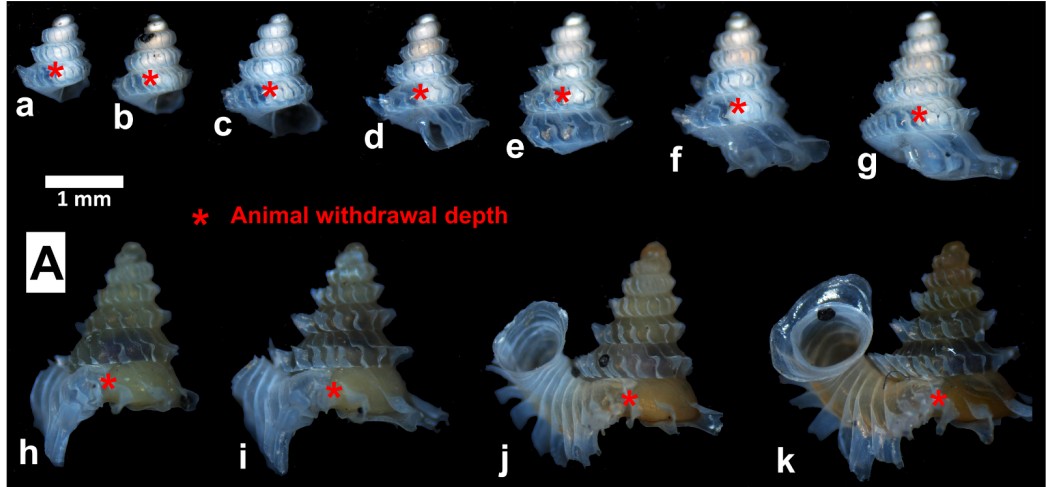

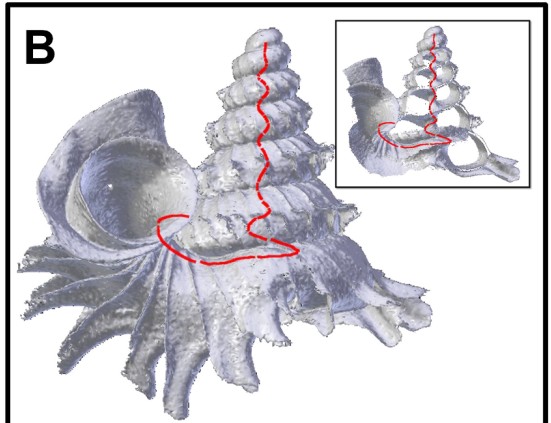

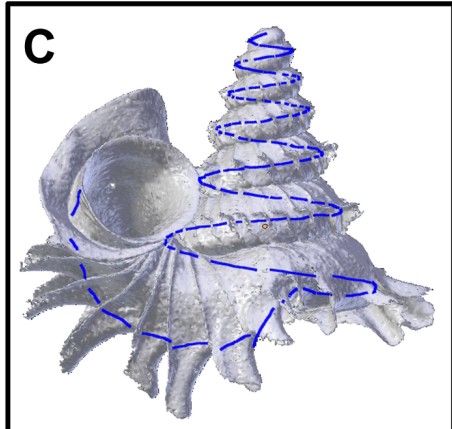

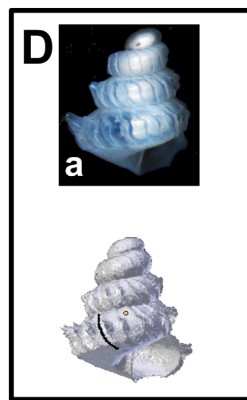

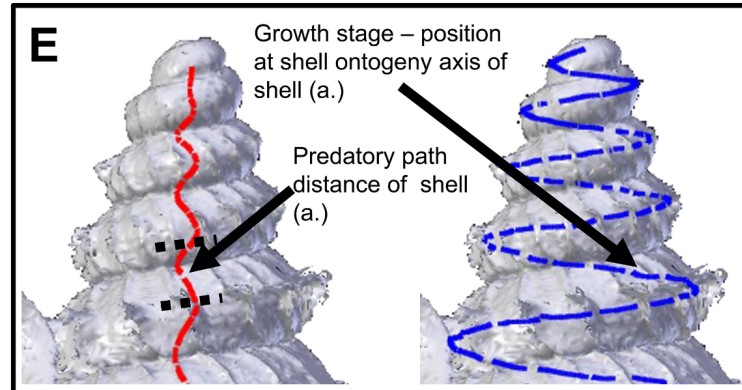

**Figure 4 Shell withdrawal path analysis of *Plectostoma concinnum* (Fulton, 1901).** (A) Animal withdrawal depth at different growth stages of the shell. (B) Predatory path in the shell (red line). (C) Shell ontogeny axis (blue line). (D) Determination of animal withdrawal depth and growth stage by using photograph and 3D shell model. (E) Transferring information of predatory path and growth stage from each shell to an adult reference shell.

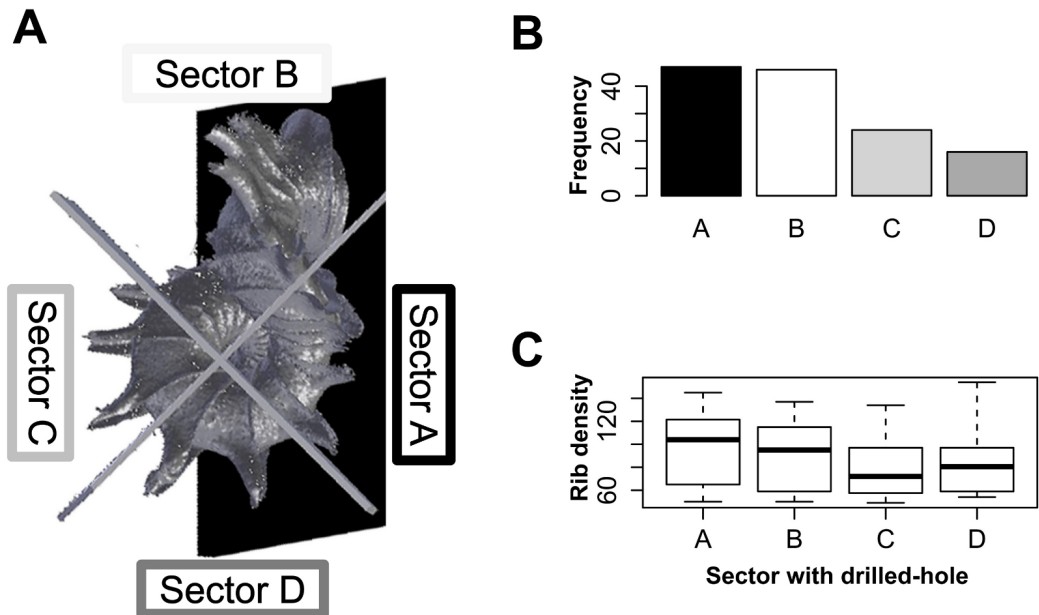

**Figure 5 Analysis of the drill hole location on the shells.** (A) Four different sectors of the shell whorls divided with reference to the snail's position when adhering to the substrate: Sector A—shell whorls facing the substrate; Sector B—shell whorls facing the tuba; Sector C—shell whorls at the back of Sector A; and Sector D—shell whorls at the back of Sector B. (B) Frequencies of drill holes found in each of four shell whorl sectors are significantly different ($\chi^2 = 22.1$, $df = 3$, $p < 0.0001$). (C) The rib density of the shells does not significantly differ among these four shell sectors (Kruskal–Wallis $\chi^2 = 3.71$, $df = 3$, $p = 0.29$).

(A) shell whorls that face the substrate; (B) shell whorls that face the tuba; (C) shell whorls opposite (A); and (D) shell whorls opposite (B) (Fig. 5A). Then, we tested if all four sectors of shell whorls are equally susceptible to slug drilling by using chi-squared test (goodness-of-fit). We also tested if the rib density (indicating prey defence), differ among these four categories with Kruskal–Wallis rank sum test (kruskal.test). All statistical analyses were done in R 2.15.1 (*R Core Team, 2012*) and R scripts can be found in File S3.

### Test 2 (c)—Effectiveness of prey's shell whorl morphometrics against shell-apertural entry by Atopos proboscis

When a *Plectostoma* snail withdraws into its shell, part of the lower shell whorls are left vacant. We named this vacant part the 'predatory path', located between shell aperture and soft-body withdrawal terminal point (i.e., between the endpoint of the shell whorls and the withdrawn snail's operculum). In shell-apertural entry predation events, the predator's feeding apparatus would need to pass through the predatory path to reach the snail that is withdrawn deeply into the shell. Hence, success of a predation event would depend on the interplay between the morphometrics of both the prey's predatory path and the predator's feeding apparatus. In this section, we quantified these morphometrics. Because both prey and predator traits vary throughout their growth, we assessed variability of these morphometrics at several different growth stages.

For the predatory path analysis, we selected from site A, 11 living snails representing a range of shell developmental stages (Fig. 4A, Specimens deposited in BOR 5656). Then, in the field, we disturbed each snail with forceps so that the animal withdrew into the shell. Immediately after that, the snail was killed with and preserved in 70% ethanol. After arriving in the laboratory, we photographed each specimen to record the withdrawal position of the animal in its translucent shell. Then, we obtained 3D models (PLY format) of these shells, based on the X-ray microtomography (µCT) technique as described in Test 1(b), using CT Analyser 1.12 (©SkyScan).

After the 3D models were obtained, we extracted the whole predatory path from the 3D model of an adult shell (hereafter "reference shell"). This is the shortest possible path when traveling inside the shell whorls from the aperture in the direction of the apex of the adult shell (Fig. 4B). We also extracted from the reference shell the whole shell ontogeny axis (*sensu Liew et al., 2014b*), which represents the entire shell's growth (Fig. 4C). Next, we determined the terminal withdrawal point for each corresponding growth stage from the photographs and 3D models of the 11 shells (Fig. 4D). After that, we calculated the distance of the portion of the whole predatory path which corresponded to the predatory path for each the 11 growth stages, and plotted these predatory path distances on the ontogeny axis (Fig. 4E). Then, we described the geometry of the shell whorls as a 3D spiral, in terms of torsion and radius of curvature (*Harary & Tal, 2011*), which were used to explore the geometry of the whorls along the predatory path.

Then, we performed the morphometrics of the slug's proboscis. However, we could not obtain an accurate measurement for the length of a fully extended proboscis because we were limited by the small number of *Atopos* specimens and the fact that the proboscis was not fully extended in most preserved specimens. Nevertheless, we attempted to estimate the length of the proboscis based on the following facts and assumptions: (1) we know that the drill hole size corresponds to proboscis diameter (*Kurozumi, 1985*; *Wu et al., 2006*); (2) we know the maximum and minimum sizes of the drill holes from Test 1(a) are 0.13 mm and 0.33 mm, which represent the range of proboscis diameters of *Atopos* in Site A and Tomanggong Besar; and (3) we assume that the dimension (i.e., diameter × length) of our slug proboscis is similar to those published for *Atopos kempii* (*Ghosh, 1913*: Plate X) (Figs. 6A and 6B). Based on this information, we estimated that the minimum and maximum dimensions of the proboscis are 0.13 × 0.8 mm and 0.33 × 1.7 mm.

Finally, we overlaid the shell predatory path with the slug proboscis morphometrics across the ontogenetic trajectory. We evaluated the growth stages for which the prey shells are not susceptible to the predator's shell-apertural entry, by comparing the morphometrics for the prey predatory path with the predator proboscis. To do this, we considered that the prey is safe from the predator when the distance of the predatory path is longer than the predator's proboscis length and when the prey's radius of curvature is smaller than predator's proboscis diameter, so that the predator's proboscis is too large to enter the shell. However, we do not know to what extent the possible exhalation or desanguination would change the proboscis diameter during the sucking.

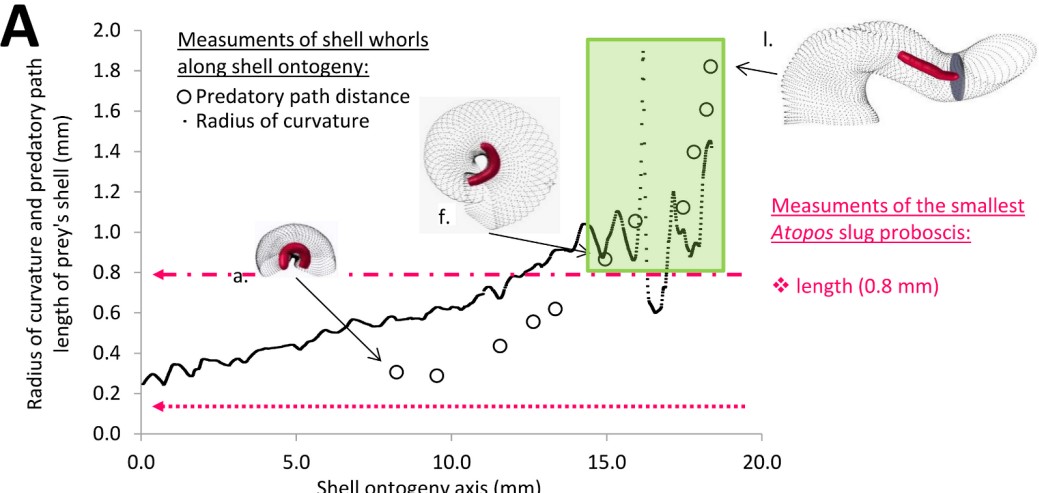

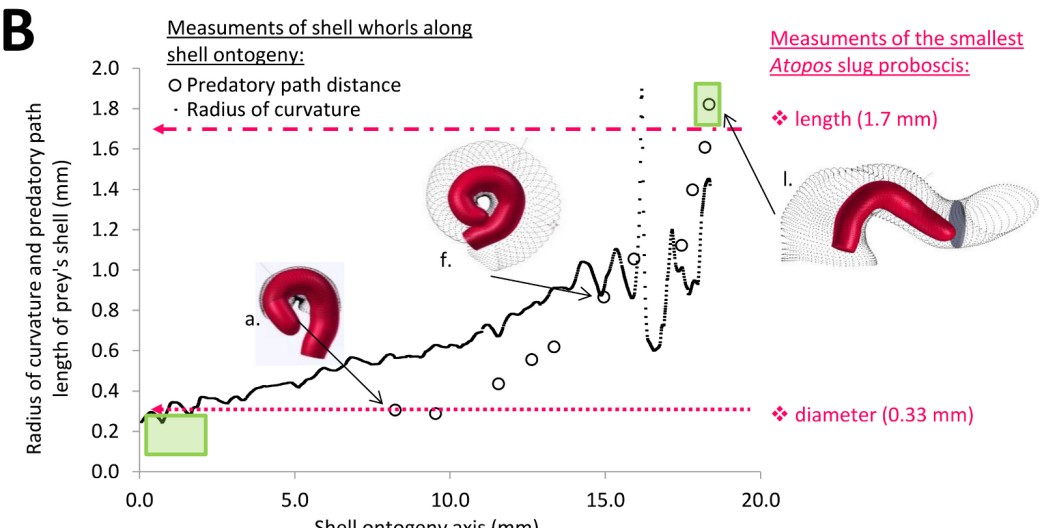

**Figure 6 Association between the predator proboscis morphometrics (pink symbols) and the prey shell whorls morphometrics (black symbols).** Green boxes represent the section of shell ontogeny (i.e., prey growth stages) that are not susceptible to *Atopos* attack by shell-apertural entry (i.e., predatory path distance > proboscis length & whorl radius of curvature < proboscis diameter). The insets show the simulation of interaction between slug proboscis and snail predatory path at three growth stages, namely, a, f and l (see Fig. 3A). (A) Smallest predator scenario. (B) Largest predator scenario.

## RESULTS

### First set of tests: (1) *Plectostoma* anti-predatory traits against *Atopos* shell-drilling behaviour

#### Test 1 (a)—Association between slug shell-drilling behaviour and adult snail shell tuba and rib density

The drill hole diameters of the 133 prey shells varied between 0.13 mm and 0.33 mm (mean = 0.230 mm, SD = 0.045, n = 133; File S2, Page 2–19: Figs. S2–S12). Four of these (3%) had two drill holes, one on the tuba and another on the spire

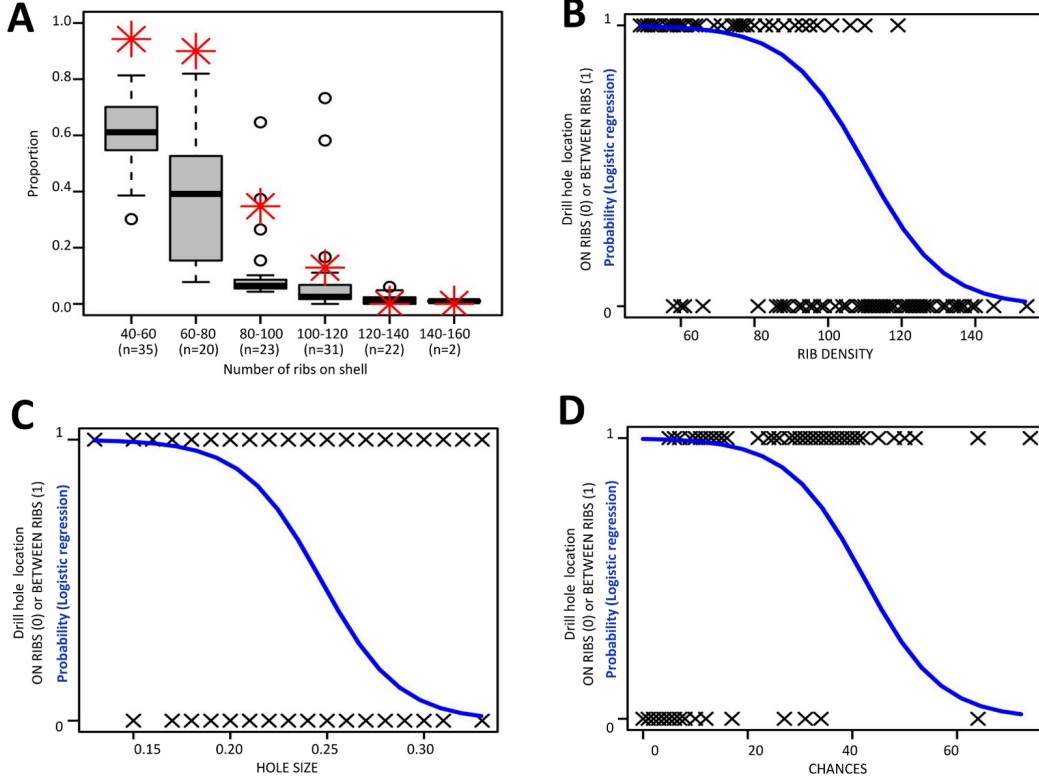

**Figure 7** **Analysis of the relationship between the likelihood of the slug drill hole BETWEEN RIBS and the three predictor variables.** (A) Proportion of the ribs spacings larger than HOLE SIZE for the shells (boxplot) and the proportion of shells having holes in between ribs (red asterisk) for each RIB DENSITY category. (B)–(D) Logistic curve showing the probability of the slug drill hole in between the ribs based on (B) RIB DENSITY (i.e., total number of ribs on shell), (C) HOLE SIZE (i.e., drill hole size, which represents the slug proboscis size), and (D) CHANCES (i.e., number of the ribs spacings that are larger than HOLE SIZE).

(File S2, page 20–21: Fig. S13). The drill hole of 70 shells (53%) was made through the ribs (ON RIBS), whereas the drill hole of the other 63 shells (47%) was made in between the ribs (BETWEEN RIBS). The result showed a logistic model that was more effective than the null model as follows: Predicted logit of (BETWEEN RIBS) = 10.448–11.316∗(HOLE SIZE) − 0.095∗(RIBS DENSITY) + 0.033∗(CHANCES), (AIC = 83.382; $\chi^2$ = 109.63, $df = 3, p = 0$; Fig. 7). According to the model, the statistically significant coefficients were for intercept ($\beta_0 = 10.448$, $Z = 2.867$, $p = 0.001$) and RIB DENSITY ($\beta_2 = -0.0916$, $p < 0.0005$; Odds Ratio = 0.91, CI = 0.87–0.95). The number of available space for drilling in between ribs (CHANCES) and the slug size (HOLE SIZE) were not significant ($p > 0.1$). In other words, the slug is less likely to drill a hole through the ribs on a densely ribbed shell, and this tendency is independent of hole size and chance.

### Test 1 (b) - Correlation between rib density and rib intensity of *Plectostoma*

Different *Plectostoma* species and populations exhibit high variability in the rib density, ranging from 49 ribs to 154 ribs per shell. There is a significant negative correlation

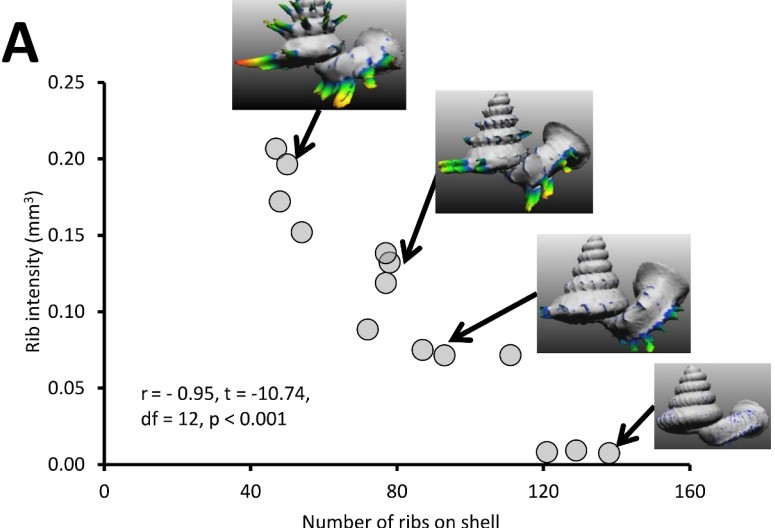

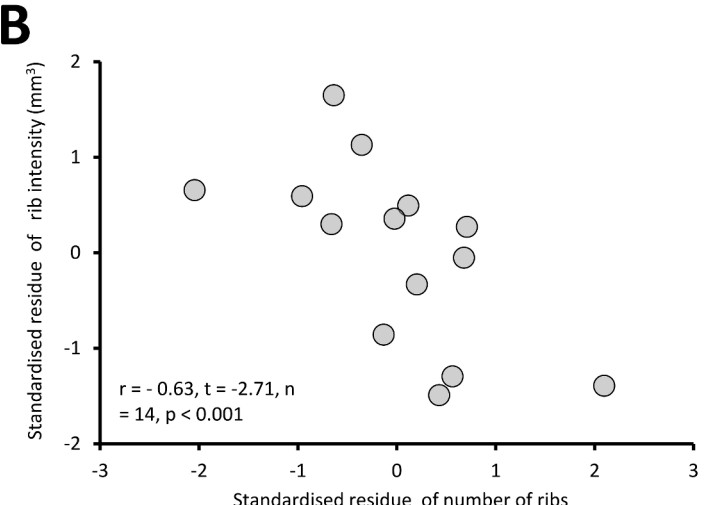

**Figure 8** **The graphs show the correlation between the number of ribs on the shell and rib intensity before and after controlling for shell size.** (A) Correlation between number of ribs on the shell and rib intensity ($r = -0.95, t = -10.74, df = 12, p < 0.001$). The rib intensity (i.e., total shell material of all shell ribs in mm$^3$) and the number of ribs were measured from 14 shells, which belong to several *Plectostoma* species and populations that vary in rib number. The inset of four examples of shells. (B) The graph shows the partial correlation of number of ribs on the shell and rib intensity after correcting for total shell material volume ($r = -0.63, t = -2.71, df = 14, p < 0.001$). The group mean values are represented by "0" on both axes.

between the rib intensity (i.e., amount of shell material in the ribs) and the number of ribs of the shell (Fig. 8A; $r = -0.95, t = -10.74, df = 12, p < 0.001$; File S2, Page 22 and 24: Table S2, Fig. S14). Both rib intensity and number of ribs are strongly correlated with the amount of shell materials after removal of the ribs (= shell size) (File S2, Page 25: Figs. S15 and S16). Nevertheless, after controlling for this, there is still a significant negative

correlation between rib intensity and number of ribs on the shell (Fig. 8B; $r = -0.63$, $t = -2.71$, $n = 14$, $p < 0.001$). These results indicate that there is a statistically significant trade-off between rib density and rib intensity, irrespective of shell size.

### Test 1 (c)—Variation of shell thickness of Plectostoma with varying shell size and number of ribs

Different *Plectostoma* populations and species have different shell thicknesses, ranging between 0.29 mm and 0.46 mm. There is a significant negative correlation between shell thickness and number of ribs (Fig. 9A; $r = -0.73$, $t = -3.70$, $df = 12$, $p < 0.005$; File S2, Page 22: Table S2). Shell thickness is strongly correlated with the amount of shell materials after removal of the ribs (= shell size) (File S2, Page 26: Fig. S17). After controlling for this, there is no significant correlation between the shell thickness and the number of ribs on the shell (Fig. 9B; $r = 0.06$, $t = -0.192$, $n = 14$, $p = 0.85$). Thus, larger *Plectostoma* shells simply are thicker.

## Second set of tests: (2) Anti-predation traits in *Plectostoma* against shell-apertural entry behaviour of *Atopos*

### Test 2 (a)—Observations on predator preference for different prey shell growth stages

Table 1 shows the snails of three ontogenetic categories that did and did not survive. It indicates that the slugs prefer to attack and consume prey with an incomplete tuba or no tuba at all (Table 1; File S2, Page 27–29: Table S4, Fig. S18). In all observations, adults with a complete tuba and peristome survived shell-apertural entry.

The predatory behaviour of the slug could not be observed directly because the slug proved very sensitive to disturbance and light. Shells of consumed prey did not show any drill-holes, which confirms *Schilthuizen & Liew*'s (*2008*) single observation that the slug attacked the juvenile prey via the shell aperture. Furthermore, 11 out of the 15 predated shells still had an intact operculum attached to the posterior side of the shell aperture (Fig. 10). It is likely that it took the slug at least seven hours to attack and consume the entire soft body of juvenile and sub-adult prey (Test no. 12 in Table 1).

### Test 2 (b)—Effectiveness of resting behaviour of Plectostoma snails against Atopos shell-apertural entry predatory behaviour

Our data show that the four sectors of the shell differ in their susceptibility to drilling by the slug (Figs. 5A and 5B; $\chi^2 = 22.1$, $df = 3$, $p < 0.0001$; File S2, Page 30: Fig. S19). Drill hole frequency is highest in sectors A and B (both 35%), and lowest in sectors C and D (18% and 12%, respectively). The high frequency of drill holes in sector A suggests that the slug is capable of removing adult prey from the substrate. Prey shell rib densities are not significantly different among the four categories (Fig. 5C; Kruskal–Wallis $\chi^2 = 7.17$, $df = 3$, $p = 0.06$), which suggests that the slug's ability to drill the hole is not influenced by the prey rib density.

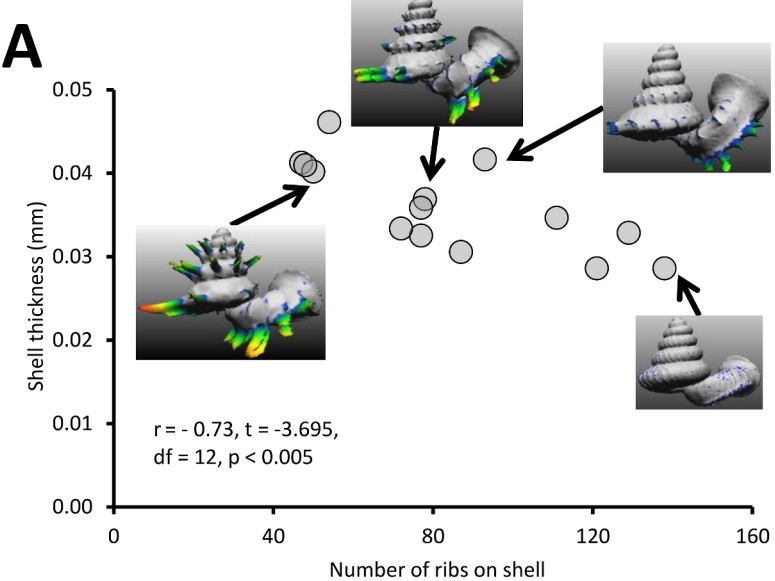

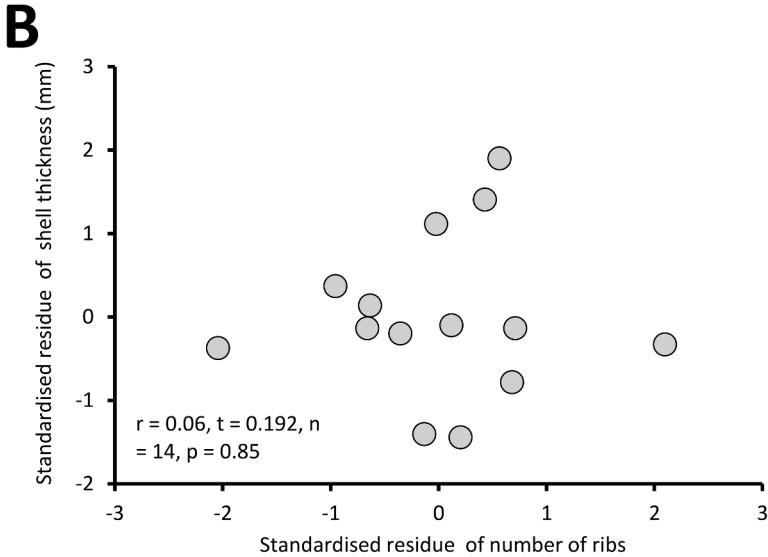

**Figure 9** **The graphs show the correlation between the number of ribs on the shell and shell thickness before and after controlling for shell size.** (A) Correlation between the number of ribs on the shell and shell thickness ($r = -0.73, t = -3.7, df = 12, p < 0.005$). The shell thickness (mm) was measured from 14 shells, which belong to several *Plectostoma* species and populations that vary in rib number. The inset of four examples of shells. (B) The graph shows the partial correlation of number of the ribs on the shell and shell thickness after correcting for total shell material volume ($r = 0.06, t = 0.19, df = 14, p = 0.85$). The group mean values are represented by "0" on both axes.

### Test 2 (c)—Effectiveness of shell morphometrics against shell-apertural entry by the Atopos proboscis

Radius of curvature (a proxy for whorl diameter) of the prey shell increases constantly with slight fluctuations throughout the shell ontogeny, apart from a few short but dramatic changes at the constriction (Figs. 6A, 6B and 11; File S2, Page 31: Fig. S20). In addition,

**Table 1 Data from Test 2(a).** Predation behaviour in relation to prey shell morphology.

| No. | *Atopos* slug ID (Table S1, File S1) | Observation starting time | Estimated starting and ending time of the predation by slug | Duration (Hour: Minutes) | Snail survivorship of each shell form category[*] | | |
|---|---|---|---|---|---|---|---|
| | | | | | **Adult** | **Sub-adult** | **Juvenile** |
| 1 | No. 7 | 22:04, 18/01/2013 | 14:00–18:30, 19/01/2013 | 4:30 | S | P[**] | P |
| 3 | No. 8 | 11:50, 20/01/2013 | 22:00, 20/01–06:00, 21/01 | 8:00 | S | p | S |
| 5 | No. 8 | 06:30, 21/01/2013 | 13:00, 21/01–22:20:00, 21/01 | 9:20 | S | p | p |
| 7 | No. 8 | 22:22, 21/01/2013 | 22:22, 21/01/2013–06:45, 22/01/2013 | 9:07 | S | p | p |
| 8 | No. 8 | 06:45, 22/01/2013 | 21:50, 22/01/2013–05:30, 23/01/2013 | 9:20 | S | p | p |
| 9 | No. 8 | 05:30, 23/01/2013 | 15:00–18:00, 23/01/2013 | 3:00 | S | p | Missing[***] |
| 10 | No. 8 | 18:15, 23/01/2013 | 18:15, 23/01/2013–10:55, 24/01/2013 | 16:40 | S | p | p |
| 11 | No. 8 | 11:00, 24/01/2013 | 18:15, 24/01/2013–09:00, 25/01/2013 | 14:45 | S | p | S |
| 12 | No. 8 | 09:00, 25/01/2013 | 23:00, 25/01/2013–06:00, 25/01/2013 | 7:00 | S | p | p |

**Notes.**

[*] "S", snail survived after experiment; "P", snail was preyed by *Atopos* slug in the experiment.

[**] Half of the animal was consumed.

[***] Specimen was lost during the handling and thus the status of survival of this individual was unknown.

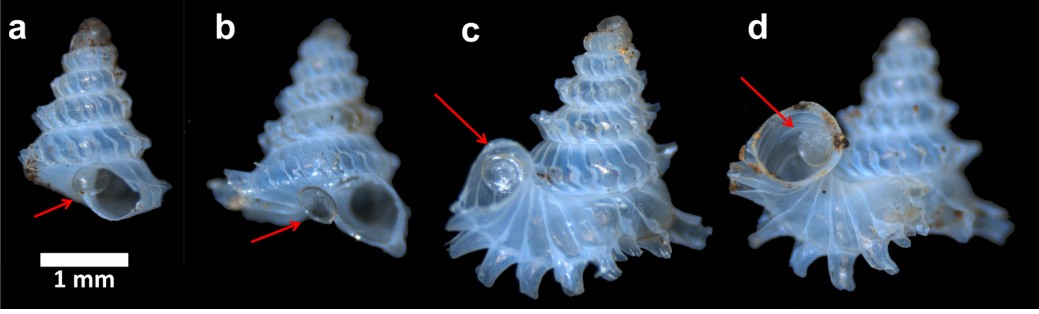

**Figure 10 Four examples of shell s after predation by apertural entry.** Each of them has an intact operculum that is attached to the posterior side of the shell aperture (arrows).

the predatory distance of the prey shell increases exponentially as the shell grows (Figs. 6A and 6B, File S2, Page 31: Fig. S21). In addition to these two morphometric changes throughout shell ontogeny, there is a dramatic change in torsion between the spire whorls and the tuba whorl (Fig. 11, File S2, Page 32: Fig. S22).

When the hypothetical slug proboscis morphometrics are plotted together with prey shell morphometrics, it becomes clear that a snail that has grown to at least five whorls would be safe from shell-apertural entry attacks by the smallest *Atopos* slug (green box in Fig. 6A). Although the slug's proboscis could fit into the whorls (proboscis diameter < radius of curvature of prey shell, Fig. 6A), it is too short to reach the soft

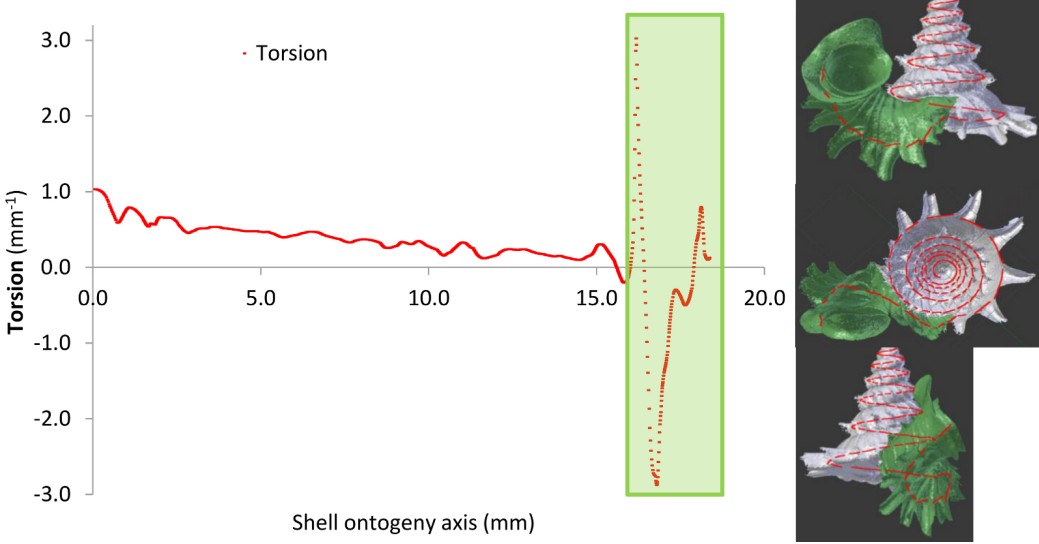

**Figure 11 Shell whorl morphometric changes in torsion along the shell ontogeny.** The tuba part undergoes dramatic changes in torsion during the shell growth.

body of an animal that has at least 5 spire whorls (slug proboscis length < predatory path distance of prey shell, Fig. 6A).

However, a larger slug could attack and consume larger prey by shell-apertural entry. A larger slug could attack prey with more than 5 spire whorls and also prey with a partial tuba because of the increase in its proboscis length and diameter (Fig. 6B). Eventually, only fully-grown prey with a complete tuba would remain safe from shell-apertural attack of a fully-grown *Atopos* slug (green box in Fig. 6B).

## DISCUSSION

### Predatory behaviour of *Atopos* slugs toward *Plectostoma* micro-landsnails

*Atopos* proved to be one of the main predators for *Plectostoma* in the two limestone hills in our small study area. Possibly, this is the case in general, because many shells of other *Plectostoma* species throughout the distribution area of the genus have the characteristic drill holes as our studied shells (Borneo, Kinabatangan region: (*Schilthuizen et al., 2006*), and Peninsular Malaysia: Liew T-S, unpublished data, 2013; File S2, Page 33–34: Fig. S23). We are not sure whether the slugs in our case are generalist predators that also feed on other snail species, as is the case with other Rathouisiidae slugs (*Heude, 1882–1890*; *Kurozumi, 1985*; *Wu et al., 2006*; *Tan & Chan, 2009*), because we have only recorded *Plectostoma* species as prey for *Atopos* in the field so far.

Predators need effective strategies to find, pursue, catch, and consume their prey (e.g., *Vermeij, 1993*; *Alcock, 1998*). Unfortunately, we were unable to study the behaviour leading up to prey attack, because we could obtain only a few live slugs, which are also very sensitive to experimental manipulation. At our two study sites, *Plectostoma* snails have high

population density (i.e., Site A, 150 individuals per m$^2$, Liew T-S, pers. obs., 18th January 2013; and Western slope of Batu Tomanggong Besar, 129 individuals per m$^2$, *Schilthuizen et al., 2003*). The abundance of *Plectostoma* snails in the vicinity of the places where *Atopos* slugs were found indicates that the slugs can easily find prey. In addition, we also suspect that the slug can effectively pursue their prey, because we observed that *Atopos* crawls faster than *Plectostoma*.

During the third stage of predation (prey capture), the prey would withdraw into the shell and adhere its shell aperture to the substrate (e.g., rock surface). The slug would attack by shell-apertural entry by removing the snail from its initial adherent position (Tests 2a & 2b), though we do not know exactly how the slug carries this out. Then, the slug holds the prey tightly in a distinctive posture (Fig. 1C, File S1, Page 1: Table S1). The same posture has also been observed in other Rathousiidae slugs (*Heude, 1882–1890*; *Kurozumi, 1985*; *Wu et al., 2006*; *Tan & Chan, 2009*). It adheres to the substrate with about two-thirds of the posterior part of the foot, and holds the prey shell with the remaining one-third, which straddles over and lays on the prey shell and pushes the shell against the substrate. On one end, the slug's head lies on the shell aperture or another part of the shell. The other end of the anterior part of the foot, which is slightly lifted from the substrate, has become thicker and might act as a pivot point. Thus, it seems to us unlikely that the snail could escape from the strong grip of *Atopos* after having been captured.

After the snail has been captured, the slug would attempt to reach the soft body by inserting its proboscis into the prey shell via the shell aperture (e.g., *Heude, 1882–1890*; *Kurozumi, 1985*; *Wu et al., 2006*; *Tan & Chan, 2009*). The slug is more likely to succeed by shell-apertural entry when the prey is not yet fully-grown (Test 2c). All other things being equal, when using the shell-apertural entry strategy, the slug would prefer to attack immature prey over prey with a fully-grown shell (Test 2a). If the slug can reach the deeply-withdrawn body of the snail (lying immediately behind the operculum) it would be able to consume it entirely (Test 2a). The slug may take more than three hours to attack and consume a juvenile snail by shell-apertural entry (Test 2a).

At the end of consumption, there is hardly any snail tissue left in the prey shell (Fig. 10). However, the operculum that had withdrawn together with the soft body into the shell remains intact and has been moved to the outside of the shell (Test 2a). We did not observe how the slug extracts the soft body from the shell, but we suppose the slug may secrete digestive fluid to dissolve the snail's tissues and then ingesting this with its proboscis, like other Rathouisiidae (*Heude, 1882–1890*; *Kurozumi, 1985*; *Wu et al., 2006*; *Tan & Chan, 2009*). Interestingly, though, these digestive fluids then do not damage the operculum (made from corneous protein) (Fig. 10; Test 2a). The operculum is free from physical damage as well.

The shell-apertural entry strategy would, however, fail if the slug's proboscis cannot reach the withdrawn soft body of snail (Test 2c; see also *Kurozumi, 1985*). In this situation, the slug uses shell-drilling to make a new opening directly on the part of the shell whorls where the snail is hiding (e.g., *Kurozumi, 1985*). We do not know how much time it takes for the slug to drill a hole on the prey shell. The holes made by the same slug individual

have the same size (File S1, Page 2) and this consistency is also known in other observations (*Kurozumi, 1985*; *Wu et al., 2006*). The exact drilling mechanism of the slug remains unknown, but it could be either mechanical or chemo-mechanical because of the narrow scraped rim on the hole margin (Figs. 1E and 1F).

The slug is able to drill holes either directly on the shell whorl surface or through the ribs (Test 1a). Nevertheless, the slug prefers to drill its hole directly on the shell surface, especially in less densely-ribbed shells, and this tendency may not simply be due to a reduced chance of hitting a rib in a shell with larger rib spacing (Test 1a, Fig. 7). Indeed, the tendency of the slug to avoid drilling holes through ribs on a less densely ribbed shell suggests that this is because ribs on a less densely ribbed shell are more "intense" (i.e., heavier; Test 1b, Fig. 8). This agrees with observations in other drilling snail predators, which also choose the thinnest part of the prey shell for attack (*Allmon, Nieh & Norris, 1990*; *Kelley & Hansen, 2003*).

In summary, *Atopos* slugs might not encounter resistance from *Plectostoma* snails during the first stages of predation. In the final stage, the slug would first attempt its shell-apertural entry strategy to insert its proboscis, and then use the alternative shell-drilling strategy if the first strategy failed. Thus, we conclude that it is likely that *Atopos* slug predation of *Plectostoma* snails is highly successful, even though the slug needs to spend more resources (e.g., time and energy) to neutralise the anti-predation shell traits of the prey. We note that *Atopos* predatory behaviour toward *Plectostoma* micro-landsnails agrees with predatory behaviours of Rathouisiidae slugs toward other snails (*Heude, 1882–1890*; *Kurozumi, 1985*; *Wu et al., 2006*; *Tan & Chan, 2009*). Hence, predatory behaviour appears to be conserved within the Rathouisiidae.

## The effectiveness of anti-predation traits of *Plectostoma* against shell-apertural entry by *Atopos*

A first line of defence of the *Plectostoma* snail against the *Atopos* slug predation is the snail's resting behaviour. When the snail is resting or disturbed, it withdraws its soft body into the shell and adheres its shell aperture firmly to the substrate. We found that the attachment of the *Plectostoma* shell aperture to the substrate may not be strong enough to resist manipulation by *Atopos*. The slug could remove the snail from the resting position and then approach the shell aperture. Hence, the resting behaviour of the snail is not an effective anti-predation trait against shell-apertural entry.

The tuba of a fully-grown shell, however, can act as a second line of defence, as it counteracts shell-apertural entry by creating a longer predatory path than the slug proboscis can traverse. However, our morphometric simulation (Figs. 6A and 6B) suggests that survival chances of juvenile snails with incomplete tuba or no tuba at all are slim under shell-apertural attack. Indeed, we have not found any drill holes on the spire of juvenile shells (Test 2a). Our estimation of the *Atopos* proboscis dimensions (i.e., length 0.8 mm–1.7 mm) agrees with those in other, similar-sized rathouissiids (*Kurozumi, 1985*: 20 mm long slug with an approximately 2-mm-long proboscis). We would like to point out that our analysis is readily re-evaluated when more data on the anatomy of *Atopos* become

available, by simply changing the threshold lines of the proboscis morphometrics in Figs. 6A and 6B (File S4).

It is worth noting that Lampyridae beetle larvae also use shell-apertural entry to attack *Plectostoma* snails. Hence, the anti-predation properties of the snail tuba against *Atopos* attack might similarly defend against the lampyrid larvae. In addition to the increased predatory path as anti-predation property, it is possible that the twisted vacant tuba whorls also help obstruct the insertion of the feeding apparatus of the slug and beetle larva if these are not flexible enough to pass through the twists of the tuba. In short, this second line of defence posed by the snail tuba could force predators to use an alternative, more costly, predatory strategy.

Open-coiled and drastic torsion of the last shell whorl like the tuba in *Plectostoma* snails has evolved several times independently in recent and extinct land and marine snails (*Vermeij, 1977*; *Gittenberger, 1996*; *Savazzi, 1996*). Such shells have a longer and less direct predatory path as compared to tightly and regularly logarithmically-coiled shells. We showed that this could be an anti-predation adaptation to shell-apertural entry by the predator (see also *Wada & Chiba, 2013*), which is opposed to the proposed association between open-coiled shell and low predation pressure (e.g., *Vermeij, 1977*; *Seuss et al., 2012*).

## The effectiveness of *Plectostoma* anti-predation traits against *Atopos* shell-drilling predatory behaviour

Upon failure of its first attempt at predation by shell-apertural entry, an *Atopos* slug will use the alternative shell-drilling strategy to consume the snail. The slug probably needs to expend more costs, in terms of time and energy, to drill a hole in the prey shell compared to the direct entry and consumption via the shell aperture. As suggested by our data (Test 2c), shell-drilling might be the only way in which *Atopos* can complete the consumption of a *Plectostoma* snail with a fully-grown shell. We did not find any signs of failed attempts of shell drilling (such as a scraped mark without a hole, or a repaired hole). Nevertheless, some of the *Plectostoma* anti-predation traits, namely, the tuba, the thickness of the shell wall, and the radial ribs could play a role in further increasing the predation cost to the shell-drilling predator.

In addition to the antipredation function towards preventing shell-apertural entry, the snail's tuba also acts as a diversionary defence against shell-drilling. When a snail has withdrawn its soft body into the spire, its tuba would be left vacant. We found evidence that the slug can be deceived, as it were, to drill a (useless) hole in the tuba (this happens rarely, though: 3% of the preyed shells in Test 1a, 8%—APO frequency in Table 1 of *Schilthuizen et al., 2006*). Moreover, the slug would then drill a second hole in the spire (Test 1a) after the first drilling attempt at the tuba. Finally, the low error rates in drilling suggests that *Atopos* individuals that frequently feed on *Plectostoma* have learned (e.g., *Kelley & Hansen, 2003*), or their populations have evolved, to distinguish the dummy tuba and the "edible" spire of the prey shell.

The penultimate line of defence against shell drilling, where shell traits are concerned, is the shell thickness. We found that shell thickness is correlated with shell size

(Test 1c, Fig. 9). Although we did not experimentally test the anti-predation role of shell thickness, we suggest that a thicker shell may not fully protect the snail from shell-drilling by the slug, because we find drill holes on the shells regardless of their shell thickness. Nevertheless, *Atopos* slugs probably need to spend more energy and time to drill a hole through a thicker prey shell.

The *Plectostoma* snail's last line of defence is the rib intensity (i.e., amount of shell material in the ribs) and rib density on the shell whorls. We found that larger shells has low rib density (fewer ribs) than smaller shells, but the ribs of the larger shells are more intense (longer and thicker) than the ribs of smaller shells. Despite the variability in rib density, all of these snails are susceptible to drilling by the slug (Test 1a, Fig. 7). Yet, *Atopos* avoids drilling through the more intense ribs on the less ribbed shells (Fig. 7).

Nonetheless, we found a trade-off between rib intensity and rib density (see next section for more discussion about this). Thus, a snail with a shell of higher rib density does not necessarily have an anti-predation advantage over a snail with a shell of lower rib density. Although we do not know if the slug would prefer prey that either have higher or lower rib density, the ribs on the prey shell do impose a greater cost for the slug because it needs to drill through these ribs before the drill hole breaches the shell wall. As suggested by *Allmon, Nieh & Norris (1990)*, the sculpture of the shell is not a very effective adaptation to resist predation by drilling. Others have suggested that tall and strong ribs could make the shell effectively larger and therefore hinder the manipulation by a predator (*Vermeij, 1977*). These hypotheses still need to be tested in the *Atopos-Plectostoma* interaction.

To sum up, *Plectostoma* anti-predation traits might mainly act to delay the predator, which increases the time and energy requirement for *Atopos* to complete predation. The resistance exhibited by the snail in response to shell-drilling by the slug cannot ensure the survival of the preyed snail. Our results are in accordance with the general view that snail shells usually cannot resist drilling by their predators (*Vermeij, 1982*).

## Why can't shell traits evolve to defend against both predatory strategies?

*Atopos* has two effective predatory strategies to neutralise the defences of *Plectostoma* during the last stage of predation. For both, it uses its digestive system (namely, its proboscis and digestive fluid in the shell-apertural entry strategy, and its proboscis, radula and digestive fluid in shell-drilling strategy). Thus, maintaining two predatory strategies that complement each other brings no additional cost to the slug development. By contrast, *Plectostoma* has to invest in two different sets of shell traits to deal with each of these predatory strategies. Yet, both sets of the shell traits have orthogonal growth directions, which indicate a possible trade-off between the shell traits.

In a hypothetical situation where predators are present that attack only by shell-apertural entry, snails can avoid predation by faster completion of a shell with a tuba, which means the snail would have to invest more resources (time and shell material) in the longitudinal growth of the shell. In the alternative situation where predators are present that attack only by shell-drilling, snails can avoid, or delay, predation by growing more

thick flaring ribs, which means it would have to invest more resources in the transverse growth and more frequent shifts from a longitudinal whorl growing mode to a transverse rib growing mode. Due to the orthogonal growth modes of these two shell traits, a snail cannot attain adult shell form faster when it needs to grow more ribs, and vice versa. This developmental trade-off causes the functional trade-off in the anti-predation traits of the shell. Therefore, none of the shell traits of *Plectostoma* are at an optimal level to defend against both shell-apertural entry and shell-drilling strategies of the *Atopos* slug.

Besides the trade-off between the two sets of shells traits, we also found a trade-off within one of these shell traits. From a theoretical point of view, the snail's shell could have evolved to have very dense, protruded and thick ribs to hinder *Atopos*'s drilling strategy. However, we found a trade-off such that ribs of more densely ribbed shells are less intense than ribs of the less densely ribbed shells. The underlying factors that cause this trade-off were not determined, but it does appear to reflect a developmental constraint.

To date, the majority of the studies of adaptive evolution of antipredation shell traits have focused on the evolution of a single shell trait of the prey in response to a single predatory behaviour of one or more predators. However, in nature, a prey might possess several antipredation traits in response to several different predatory behaviours of a predator (e.g., *Sih, Englund & Wooster, 1998*; *DeWitt & Langerhans, 2003*; *Relyea, 2003*). Usually, a snail will counteract a particular predatory strategy with a single evolved anti-predation shell trait (*Vermeij, 1993*), but snails sometimes use a combination of more than one trait to defend against a predatory strategy (*DeWitt, Sih & Hucko, 1999*; *Wada & Chiba, 2013*). A few studies have shown that there may be a functional trade-off between such multiple anti-predation traits. For example, *Hoso (2012)* demonstrated that two snail anti-predation traits evolved by changes in two different developmental mechanisms (shell coiling direction and foot structure) in response to two predation stages (capture and consumption) of the same predator. Here, we show another novel context of an anti-predation functional trade-off between two sets of anti-predation shell traits that are part of the same developmental mechanism (shell ontogeny), but in response to two different predatory behaviours within the same predation stage (consumption) by the same predator.

We found several correlations and trade-offs between and within the sets of anti-predation shell traits with each set having a specific function against a particular predatory strategy. However, more study is needed to clarify the exact causal relationships and to determine the underlying developmental biology of these shell anti-predatory traits. This could have important implications for our understanding of the evolutionary adaptability of shells under predation selection pressure in *Plectostoma* snail in particular and Gastropoda in general.

## CONCLUSION

Our study has unravelled several aspects of the predator–prey interactions between the *Atopos* slug and *Plectostoma* snails in the limestone habitats of Borneo. Despite having several distinct anti-predation traits, such as protruding radial ribs and distorted coiling of

the shell, *Plectostoma* snails have low resistance against predation by the slug with its two predatory strategies (shell-apertural entry and shell-drilling). Lastly, the effectiveness of the snail's anti-predation traits is probably limited by trade-offs imposed by ontogenetic constraints.

## ACKNOWLEDGEMENTS

We are thankful to Effendi bin Marzuki, Heike Kappes, Angelique van Til, Mohd. Sobrin, and Samsudin's family for their assistance in the fieldwork. We are grateful to Willem Renema for introducing LTS to CT-Scan instrumentation. Finally, we would like to acknowledge Thomas DeWitt, Dany Garant and Scott Large for providing useful comments that improved the manuscript.

### Funding

This study is funded under project 819.01.012 of the Research Council for Earth and Life Sciences (ALW-NWO). The funders had no role in study design, data collection and analysis, decision to publish, or preparation of the manuscript.

### Grant Disclosures

The following grant information was disclosed by the authors:
Project 819.01.012 of the Research Council for Earth and Life Sciences (ALW-NWO), The Netherlands.

### Competing Interests

The authors declare that they have no competing interests.

### Author Contributions

- Thor-Seng Liew conceived and designed the experiments, performed the experiments, analyzed the data, contributed reagents/materials/analysis tools, wrote the paper, prepared figures and/or tables, reviewed drafts of the paper.
- Menno Schilthuizen contributed reagents/materials/analysis tools, wrote the paper, reviewed drafts of the paper.

### Field Study Permissions

The following information was supplied relating to ethical approvals (i.e., approving body and any reference numbers):

The permissions for the work in the study sites were given by the Wildlife Department of Sabah (JHL.600-6/1 JLD.6, JHL.6000.6/1/2 JLD.8) and the Economic Planning Unit, Malaysia (UPE: 40/200/19/2524).

### Data Deposition

The following information was supplied regarding the deposition of related data:
http://dx.doi.org/10.6084/m9.figshare.830399.

## Supplemental Information

Supplemental information for this article can be found online at http://dx.doi.org/ 10.7717/peerj.329.

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
