# Peer review of "Association between shell morphology of micro-land snails (genus Plectostoma) and their predator’s predatory behaviour"

_PeerJ, doi:10.7717/peerj.329_

## Round 0.1 · original submission · Major Revisions

After a slight delay, we have now received two reviews on your manuscript. Both reviewers found merits in your study, but also raised important points that deserve further attention. First, your manuscript greatly needs to be streamlined and some sections could easily be removed; the structure also needs to be carefully revised to improve the focus of the paper. At the same time, some relevant literature has been overlooked and needs to be integrated to the manuscript (the reviewers made a few suggestions in that respect but a thorough review of the literature should also be performed). Importantly, both reviewers emphasized that the statistical tests performed should be better explained. As they stand, they are confusing and one cannot conclude on the validity of the findings, which is quite worrying. Also, the hypotheses and predictions should be clarified in a single section at the end of the introduction. Finally, the reviewers provided several additional points that need to be assessed in your revision to further improve your manuscript. As such, you should present a detailed cover letter that will describe your response to every comment.

·

Basic reporting

Overkill. The paper I feel suffers from reporting every detail instead of focusing on summative commentary. But this is stylistic and for this forum may be acceptable.

Experimental design

Often weak, but in such instances manageable with cautious language.

Validity of the findings

Great. Important, relevant and solid data of significant interest to the behavioral and evolutionary ecology fields

Additional comments

Review of “Association between shell morphology of micro-land snails (genus Plectostoma) and their predator’s predatory behavior” by P Liew et al, manuscript for PeerJ.

This paper is a mosaic of good and bad practices. I would like to see it published in some form as the data are valuable and worthwhile. The Abstract and Introduction are admirably composed, with thorough scholarship, but then things get very inefficient, at best. The manuscript reports great detail with too little of the distillation we typically see in scientific papers. My biggest problem with the paper is the ‘report-every-detail’ approach. For example, they tabulated (placed in Tables) each observation they found relevant in their literature review, AND each predator-prey interaction they observed in the study. Other oddities occur as well, such as placement of the literature review within the experimental procedures. Literature review in scientific writing is typically integrated into the Introduction and Discussion sections—not placed in Tables or treated in the Methods or Results.

The writing is generally clean within sentences but often suffers poor architecture (inter-relation of adjacent sentences, paragraphs, etc.). For example the first paragraph of the Discussion:
“In general, our results show that in attacking and consuming the unusually-shaped _Plectostoma, the slug Atopos uses the same predatory strategies that are widespread in other members of the slug family Rathouisiidae. The Atopos population in this study was found on humid and shaded limestone rock surfaces._ In suitable habitat, up to 15 slugs could be found in 25 m2 of rock face
(no. 1 in Table 1). The slug is a nocturnal predator and it was seen foraging at night and, in shady places, also early in the morning. During the day, the slug probably hides in the cracks of the limestone rock. Similar ecological characteristics have been reported for other Rathouisiidae.” Odd mix of information and detail either too much or in odd position. The two underlined sections are entirely disconnected thoughts. So, the first paragraph of the Discussion, among the most important areas of text for such a paper, is simply illogical.

***This paper needs to be streamlined*** as per the usual standards of scientific writing (publication of a much reduced manuscript is warranted). This manuscript is currently 49 manuscript pages single spaced! 34 pages of supplementary material!

I also find the ***discussion of Red queen to be pointless*** as this paper did not document any evolutionary response of the predators to the prey. The last section of the paper, previous to the concluding paragraph, could be removed.

***So in summary this paper is atypical of scientific writing in behavioral and evolutionary ecology. But I want to be clear that there is value here***. The behavior and ecology of the system is interesting. The data are important. The figures are spectacular. The analysis, I have some concerns about as addressed in comments below. A major revision may be publishable.


Line by line comments:
Line 24: “Among the studied prey traits, those of snail shells, which act like armours,” awkward usage of word armours.

“Since Goodfriend’s (1986) review, few additional studies have shown the adaptive significance of land snail shell traits under predation pressure, namely, aperture form by Gittenberger (1996), Quensen and Woodruff (1997), Hoso (2012) and Wada and Chiba (2013); shell form by Quensen and Woodruff (1997), Schilthuizen et al. (2006), Moreno-Rueda (2009) and Olson and Hearty (2010); shell ribs by Quensen and Woodruff (1997); and shell coiling direction by Hoso and Hori (2008).” be sure to look at Konuba & Chiba, I think in American Naturalist.

line 105: “… radial rib density and intensity.” What is rib intensity?

line 114: “To date, only one direct field observation of shell-drilling by Atoposis available…” Is available.

Test 1 (a) – Atopos drill hole characteristics on the shell of adult Plectostoma.
Huge shame the authors did not compare living Plectostoma to those with drill holes or dead shells lacking holes. Though even so, lacking replication this was a weak test to draw any inferences.

Test 1 (b) – a
“Conversely, if snail shell ribs are adaptive traits in the context of the slug’s shell-drilling behaviour, we would expect the snail shell to have evolved more densely-placed, thicker, and more protruded ribs.” More densly-placed, thicker and more protruded compared to what? Without a comparison it is hard to know what authors are going for here.

line 253: “These three preys…”

line 267: “summed up” summed.


Test 2 (a) – heuristic only statistical test—observations not independent. 24 of 27 observations from one predator.

line 278: “the sector of the shell facing the substrate” meaning aperture?

Test 2 (b) seems fine

line 332: “preyis”

we considered that preyis safe from the
predator when the distance of the predatory path is longer than the predator’s proboscis length
and when prey’s radius of curvature is smaller than predator’s proboscis diameter, so that
predator’s proboscis is too large to enter the shell.” be aware this will be an underestimate of the threshold – due to the potential for exhalation or desanguination

line 364: “slugand”

Fig S15 “Shell materials of shell whorls” ? what does that mean? appears elsewhere in online tables

Table S3. What does 0 mean?

Fig S18: “preyed specimen”
“Specimen lost during the
photograph (no picture)” What does mean?

Figure S23. “Examples of other Plectostoma species that found in…”


Line 376: “The number of ribs of the six shell vary…”

Fig S18: “preyed by Atopos slug” … “preyed specimen”. Fix grammar.

Line 492: “which is slightly lifted from the substrate, has becoming thicker”

Disturbingly little citation in Discussion at least through line 518.

line 527: “is because ribs on a less densely ribbed shell are more “intense” (i.e., heavier; Test 1c, Figure 6).” I still don’t get it; what does intense mean in this context?

Line 535: need some citations to the increased handling time stuff—maybe classic mussel OFT work, my work on handling time in freshwater snails (http://people.tamu.edu/~tdewitt/DeWitt,%20Robinson,%20Wilson%20(2000)%20Evol%20Ecol%20Res.pdf), etc. By the way, my W should be capitalized (line 769) .

Line 540: “The first line of defence of the Plectostoma snail against the Atopos slug predation is the snail
resting behaviour.” ‘The’ first line is habitat choice. Can revise to indicated ‘A’ first line…

Line 566: “Such shells [having a tuba] have a longer predatory path…” Longer and less direct.

Line 592: “The penultimate line of defence against shell drilling, where shell traits are concerned, is the shell thickness.” penultimate for these snails perhaps—other snails have further defenses like desanguination (see Jukka Jokela’s work).

Line 597: Again, my paper in 2000 very relevant for this discussion.

Line 644: “However, in nature, a prey might possess several antipredation traits in response to several different predatory behaviours of a predator.” A vital citation on this point and more generally to support much else of the authors’ arguments, especially selection on multiple antipredator traits in snails, is my 2003 JSR paper: http://people.tamu.edu/~tdewitt/DeWitt,%20Langerhans%20(2003)%20J%20Sea%20Res.pdf

Line 675: “Schilthuizen et al. (2006) examined drill hole patterns for 16 populations of Plectostomato
establish possible links…” broken sentence.

Line 679: “(represented by principal component scores calculated from logtransformed
linear measurements of shells)” unnecessary—delete.

Line 686: “was genetically determined and modulation by shell morphology.” broken

The whole discussion of red queen feels inappropriate as this paper doesn’t address evolution of the predators. Even the discussion of Schilthuizen et al. (2006) does not seem to indicate, though the language is confusing as noted above (for line 686), that they studied the predator’s evolution either.

Paper is WAY too long.

Reviewer 2 ·

Basic reporting

The article presents an interesting investigation into the morphology and suitability of land snail armament. The author's present interesting techniques that utilize x-ray microtomography as well as laboratory experiments to address how shell traits, namely the tuba and shell ribs, may be used to avoid predation. The authors include a flowchart of included tests and questions, however, these have not been culled and formulated into a coherent structure. The materials and methods section is a laundry list of tests, which lack a unifying focus and detracts from the more interesting and relevant discoveries. The reader is left to sift through Although R script is included, it does not include

Specifics:

Line 38- is there a citation for this?
Line 51- Awkward sentence. Embed citations and reword.
Line 61- Typically Figure 1 proceeds Figure 2. Renumber. In general, double check figures with citations within text, as many were incorrect.
Line 74- (And title) predator's predatory behavior... awkward at best. Please reword this phrase.
Line 76- Reword and focus into CLEAR hypotheses. Right now you have several that are very scattered. You have very interesting techniques and findings that need to be communicated, but the structure is very weak.
Lines 94-112- The literature review and predation tests should be reworded and included in the introduction. These were initial findings that you made that helped you formulate hypotheses and direction for the rest of the experiments. Also, I don't see the need to introduce an experiment examining Lampyriid predation strategies when you actually could not conduct the experiment.

Lines 459- 474 provide important background that help introduce the questions... I suggest you visit this in the introduction.

Line 467- evidence? citation?

Experimental design

Test 1(a): No replication. Perhaps include this in introduction as "... preliminary findings suggest that Atropos drill holes are distinctive and represent proboscis size..." however, I'm not certain you actually tested that claim since you have a sample size of 1 predator. Also, the slug died? This is not a test, these are observations. Tests have been replicated.
Test 1(b): You collected shells that had a characteristic Atropos hole? Is this based upon the characteristics from the (single) slug in Test 1(a)? How are you certain that these characters are actually representative of Atropos, not of the single slug you examined? You are supporting many claims with observations, which is fine, but don't suggest that they correspond to population characteristics. Also, you don't include your logistic regression model in the R script.
Test 1(c): What is the accuracy of the measurements? Should be reported.
Test 1(d): How accurately can you measure the thickness? Are differences of 0.01 mm actually meaningful and measurable?

Test 2(a): what is the replication here? You only used 2 (well, 1 really...) slugs? Are these actually independent events? Line 268- the slug ate the sub-adult and juvenile snails... can you actually claim that the likelihood of predation is equal between all three groups?
Test 2(b): Do you have enough power to make this assertion? You need to do a power analysis here. Also, you can't say that slugs of all ages and sizes are capable of manipulating prey... your claim that proboscis size is representative of slug size is untested! You have n = 1 defining "characteristic" Atropos holes and claim that slugs of all sizes can manipulate prey. It's like claiming that since Kobe Bryant can slam dunk a basketball so can Margaret Thatcher... If the references for Line 322 bolster this argument, why don't you discuss and introduce much earlier.

Test 2(c): Probably most interesting component, but again you have not fully convinced me that proboscis length scales with drill hole size. Also, are there coefficients of variation or other forms of significance that can be applied? Really, you don't validate your findings. You are plotting an estimated proboscis length and snail morphometrics and making rather broad assumptions...

Figure 7(a) scale is either not displaying enough significant figures or is incorrect.

Validity of the findings

The findings are interesting, however, I am not fully convinced that the statistics are entirely appropriate and that suitable replicates have been carried out. I realize that these organisms are probably difficult to manipulate and collect, but the reader should be instructed of this. I don't believe the chi-square (goodness of fit) tests are entirely appropriate. Do you have expected outcomes that inform this statistic? Perhaps I am missing something here...

Additional comments

These are interesting ideas and an interesting study, but I think you need to refocus your efforts to make a more concise and compelling study. You propose many, many interesting ideas that are only partially supported with findings. I agree that the findings are important and relevant, but I am not fully convinced (statistically) of the validity of your claims. I encourage you to revisit this and to provide a more rigorous investigation.

---

## Round 0.2 · accepted · Accept

Thank you for your revisions. The manuscript is now acceptable for publication in PeerJ.

·

Basic reporting

I find the revised version of the manuscript much easier to comprehend, digest, and enjoy. Thank you for working hard to cull the extra detail into the supplementary sections. I believe that further simplification could be achieved, but what you have accomplished is sufficient.

Experimental design

Thank you for addressing my comments regarding the experimental design and re-focusing the manuscript. The micro x-ray technique is very interesting and I foresee further applications will abound. Also, thank you for including the updated R script.

Validity of the findings

Test 2(b) line 379: Please be very careful not to give p-values too much importance... having a very low p-value (p < 0.0001) does not necessarily indicate that the size of the effect is very different from a high p-value (p<0.06), especially with such low sample sizes.

Additional comments

Thank you for working hard to revise this manuscript. Best of luck.